# Solvent Deuterium Oxide Isotope Effects on the Reactions of Organophosphorylated Acetylcholinesterase [note 1]

**DOI:** 10.3390/molecules25194412

**Published:** 2020-09-25

**Authors:** Terrone L. Rosenberry

**Affiliations:** Mayo Clinic College of Medicine, Departments of Neuroscience and Pharmacology, Jacksonville, FL 32224, USA; rosenberry@mayo.edu ; Tel.: +1-904-953-7375; Fax: +1-904-953-7370

**Keywords:** acetylcholinesterase, paraoxon, dephosphorylation, reactivation, D_2_O isotope

## Abstract

Organophosphates (OPs) are esters of substituted phosphates, phosphonates or phosphoramidates that react with acetylcholinesterase (AChE) by initially transferring the organophosphityl group to a serine residue in the enzyme active site, concomitant with loss of an alcohol or halide leaving group. With substituted phosphates, this transfer is followed by relatively slow hydrolysis of the organophosphoryl AChE, or dephosphorylation, that is often accompanied by an aging reaction that renders the enzyme irreversibly inactivated. Aging is a dealkylation that converts the phosphate triester to a diester. OPs are very effective AChE inhibitors and have been developed as insecticides and chemical warfare agents. We examined three reactions of two organophosphoryl AChEs, dimethyl- and diethylphosphorylated AChE, by comparing rate constants and solvent deuterium oxide isotope effects for hydrolysis, aging and oxime reactivation with pralidoxime (2-PAM). Our study was motivated (1) by a published x-ray crystal structure of diethylphosphorylated AChE, which showed severe distortion of the active site that was restored by the binding of pralidoxime, and (2) by published isotope effects for decarbamoylation that decreased from 2.8 for *N*-monomethylcarbamoyl AChE to 1.1 for *N*,*N*-diethylcarbamoyl AChE. We previously reconciled these results by proposing a shift in the rate-limiting step from proton transfer for the small carbamoyl group to a likely conformational change in the distorted active site of the large carbamoyl enzyme. This proposal was tested but was not supported in this report. The smaller dimethylphosphoryl AChE and the larger diethylphosphoryl AChE gave similar isotope effects for both oxime reactivation and hydrolysis, and the isotope effect values of about two indicated that proton transfer was rate limiting for both reactions.

## 1. Introduction

Acetylcholinesterase (AChE) catalyzes the hydrolysis of the neurotransmitter acetylcholine, and rapid hydrolysis of this ester is essential for normal cholinergic synaptic transmission. Acetylcholine hydrolysis proceeds by transfer of the acetyl group to the active site serine of AChE followed by hydrolysis, or deacetylation, of the acetyl enzyme, and both steps occur on a timescale of microseconds [1]. Other ester substrates of AChE proceed through a similar two-step catalytic pathway (Scheme 1). However, the pathways for the esters in Scheme 1 differ from that of acetylcholine in that these esters initially form a detectable reversible equilibrium complex with AChE with a dissociation constant, *K*_D,_ before transfer of their carbamoyl or organophosphoryl groups (here referred to as acyl groups) to the active site serine to give carbamoyl or phosphoryl intermediates. This complex is an intermediate with acetylthiocholine but is not detectable, as explained in Equation (1) below. While the hydrolysis of the carbamoyl or organophosphoryl intermediates involves transfer of their acyl group from serine to water, their deacylation rate constants (*k*_3_ or *k*_H_) are orders of magnitude smaller than that of acetylated AChE [2,3]. Their slow deacylation makes them potent inhibitors of AChE, and some carbamates and organophosphates (also denoted OPs) have been widely used as insecticides [4].

One noteworthy feature of AChE intermediates with carbamoyl and phosphoryl acyl groups (Figure 1) is that their relative deacylation rate constants *k*_3_ or *k*_H_ decrease dramatically with an increase in acyl group size. This decrease may result in part from substantial distortion of the active site arising from the bulk of large acyl groups [8]. For example, an X-ray crystal structure of diethylphosphorylated hAChE generated from paraoxon [9] is shown in Figure 2A. In this structure, the diethylphosphoryl group is attached to Ser203. One ethoxy group faces Trp86 and the other faces Phe295, Phe297 and Phe338 in an acyl pocket. Hydrogen bonding within the catalytic triad (Ser203-His447-Glu334) remains intact but the adjacent acyl pocket residues and acyl loop (residue positions 280–297) are significantly perturbed relative to their positions in the ligand-free hAChE structure. Movements in residue backbone positions are necessary to avert steric clash, and the Cα and the sidechain of Arg296 shift by 4.9 A° and 14.9 A°, respectively. Large acyl loop backbone rearrangements have also been seen in mAChE and TcAChE inhibited by the OP diisopropylfluorophosphate [10,11]. However, stereoselective inhibition of AChE by OP nerve agents that place a smaller methyl group into the acyl pocket does not affect the acyl loop in the crystal structure in the same manner [9,10,12].

These shifts in the acyl pocket residues and acyl loop in diethylphosphorylated hAChE can be largely reversed by cationic ligand binding to the active site [9]. The interaction of cationic oximes with organophosphorylated AChEs is of intense interest because oximes have been shown to reactivate these AChEs by displacing the serine hydroxyl and releasing the organophosphorylated oxime [7,13]. The structure in which 50 mM 2-PAM was diffused into the crystals to form a complex of diethylphosphorylated hAChE with the cationic oxime 2-PAM reveals two bound 2-PAM molecules (Figure 2B), one in the peripheral site and the other stacked against the sidechain of Trp86. In this ternary complex, the backbone conformation of the acyl loop resembles that of the ligand-free state and appears to be stabilized by oxime binding. Therefore, diffusion of these oximes into the crystals of diethylphosphorylated hAChE promoted a restoration of the acyl pocket residues and acyl loop to their locations in the unmodified hAChE structure.

The catalytic effects of active site distortion in diethylphosphorylated AChE on the deacylation pathway can be examined by determining D_2_O isotope effects on the deacylation reactions. Here we investigate the reaction rate constants and D_2_O isotope effects for three reactions of diethylphosphorylated AChE, which shows crystallographic evidence of active site distortion, and of dimethylphosphorylated AChE, which does not [9,10,12].

## 2. Results

The primary objectives of this paper are to obtain the solvent D_2_O isotope effects for three of four reactions that involve organophosphorylated AChEs. The four reactions are outlined in Scheme 2, and the isotope effects are shown in Table 1 below. Because the determination of solvent D_2_O isotope effects requires accuracy and precision, an emphasis is placed on protocols for rate constant measurement. The isotope effects obtained are then compared to those previously measured for carbamoylated AChEs, and differences among them are interpreted in the context of X-ray crystal structures determined for organophosphorylated AChEs like those shown in Figure 2.

Protocols to measure the rate constants of the reactions of organophosphorylated AChEs in Scheme 2, in both H_2_O and D_2_O, are outlined in the Materials and Methods, and representative reactions are illustrated in Figure 3, Figure 4 and Figure 5 below.

### 2.1. Denaturation Rate Constants k_X_ for Organophosphorylated AChE

Before the rate constants for reactions b, c and d in Scheme 2 can be determined, the loss of AChE activity arising from enzyme denaturation must be taken into consideration. The reaction denoted denaturation in Scheme 2a was detected with control AChE and with fully reactivated AChE in the absence of OPs. Bovine serum albumen (BSA) was added to the assay solution to stabilize the AChE, but the enzyme still showed a slow loss of activity with a rate constant *k*_X_ that (1) in some cases was close enough to the aging rate constant *k*_A_ and the hydrolysis rate constant *k*_H_ to require inclusion in the analytical equations and (2) was stabilized by 2-PAM and thus depended on the concentration of 2-PAM. Individual measurements of *k*_X_ ranged from 2–7 × 10^−5^ min^−1^ and are illustrated in Figure 4 and Figure 5A below. The same denaturation constant *k*_X_ was assumed to apply to free and organophosphorylated AChE.

### 2.2. Aging Rate Constants k_A_ for Organophosphorylated AChE

The fact that organophosphorylated AChE can undergo multiple reactions (Scheme 2) required a sequential approach to obtaining the rate constants for these reactions. Fitting some of the data here to equations required preliminary estimates of some rate constants that were refined in the fitting of later data. The aging reaction in Scheme 2b is the loss of an alkoxy group attached to the phosphorus atom [14]. The resulting phosphate diester that remains covalently linked to AChE is an extremely stable inactive enzyme form that cannot be reactivated by oxime (Figure 3). The protocol we used to measure the aging rate constants *k*_A_ is indicated in the Figure 3 legend. We followed this protocol to obtain our best estimates of *k*_A_ in both H_2_O and D_2_O. AChE was maintained in a fully organophosphorylated form for various times *t*_aged_, after which it was diluted into a high concentration of 2-PAM to rapidly reactivate all of the enzyme that had not been aged. Figure 3A demonstrates that, with reasonable initial estimates of *k*_X_, *k*_A_, *k*_H,_ and *k*_r_, 2-PAM reactivation proceeds rapidly. Fitting confirmed that all values of *v* measured after the initial value near *t* = 0 essentially corresponded to the final value of *v*, denoted *v*_F_. Plots of these *v*_F_ against *t*_aged_ could be analyzed with Equation (3) as shown in Figure 3B to give values of *k*_A_ presented in Table 1.

### 2.3. Hydrolysis Rate Constants k_H_ for Organophosphorylated AChE

The hydrolysis rate constants *k*_H_ were measured in H_2_O and D_2_O with the protocol in the Figure 4 legend. AChE was organophosphorylated and diluted with or without 200 μM 2-PAM. Assay values *v* from the dilution with 2-PAM rapidly reached maximums and were analyzed with Equation (2) to determine *E*_tot_. This value of *E*_tot_ was then fixed in Equation (2) to fit the values of *v* from the dilution without 2-PAM to obtain *k*_H_ (Figure 4).

In contrast to the protocol for the aging reaction in Figure 3A, where a slight excess of OP during preincubation was rapidly turned over by dilution into a high concentration of 2-PAM, the hydrolysis reaction was defined in the absence of 2-PAM and its time course was sensitive to any excess of OP. An excess was indicated by an *E*_0_ close to zero and by a lag in the increase in *v* with time. Centrifugation through a spin column was employed to minimize unreacted OP but, in addition, the stoichiometry of AChE and OP was set close to one and the incubation time was limited to insure that, in most cases, *E*_0_ was significantly greater than zero. However, this was not the case in Figure 4, where *E*_0_ approached zero.

### 2.4. 2-PAM Reactivation Rate Constants k_R_ for Organophosphorylated AChE

The reactivation rate constants *k*_r_ depend on the concentration of 2-PAM, but at the 2-PAM concentrations employed here they were considerably larger than rate constants for denaturation, aging and hydrolysis described above. Nevertheless, the determination of accurate solvent D_2_O isotope effects required inclusion of these smaller rate constants in Equation (2). The protocol for obtaining values of *k*_r_ is shown in Figure 5A. A rapid increase in assay values *v* was observed on adding organophosphorylated AChE to 2-PAM, followed by a much slower decrease in *v* that we attributed to enzyme denaturation. The much longer slow decrease dominated the fitting procedure when the entire time course was analyzed, so fitting was divided into two steps. In the first step only the denaturation rate constant *k*_X_ was retained, and it showed a clear decrease in value at higher 2-PAM concentrations (Figure 5A) as noted above. The second fitting step retained the value of *k*_X_ from the first step as an additional fixed variable, and *k*_r_ was fitted from the initial increase in *v* (Figure 5A). Values of *k*_r_ obtained at different 2-PAM concentrations were then analyzed according to Equation (4) to obtain the maximum reactivation rate constant *k*_R_ at concentrations of 2-PAM that saturated the organophosphorylated enzyme (Figure 5B).

### 2.5. Accuracy of Rate Constants for Reactions of Organophosphorylated hAChE

In addition to the rate constants given in Table 1, we also determined *k*_X_ values in the absence of 2-PAM in H_2_O ((53 ± 7) × 10^−6^ min^−1^, *n* = 7) and in D_2_O (41 ± 14) × 10^−6^ min^−1^, *n* = 3). These values are significant because, in some cases, several measured rate constants are close in value, rendering the accuracy of one rate constant highly dependent upon another. For example, values of *k*_H_, *k*_A_ and *k*_X_ for diethylphosphorylated AChE are almost identical in D_2_O. The rate constants most sensitive to these equivalencies are the *k*_A_ values in Table 1. These values were obtained by fitting Equation (3), where the exponential rate constant is *k*_A_ + *k*_X_. Therefore, from the 40 × 10^−6^ min^−1^ value of *k*_A_ in Table 1, a 2-fold increase in *k*_X_ renders *k*_A_ close to zero. The value of *k*_X_ also plays an important, although lesser, role in the determination of *k*_H_ for diethylphosphorylated AChE in D_2_O (Table 1 and Figure 4): a 2-fold increase in the fixed value of *k*_X_ in Equation (2) increases *k*_H_ by 45%, whereas a 2-fold decrease in *k*_X_ decreases *k*_H_ by 20%. Values of *k*_H_, *k*_A_, and *k*_X_ for dimethylphosphorylated AChE diverge to a much greater extent (see Table 1), making their accuracy largely independent of each other. In view of these differences in the rate constant interdependence of *k*_A_ and *k*_H_, the similarities in the solvent D_2_O isotope effects for dimethyl- and diethylphosphorylated AChE in Table 1 are reassuring but possibly serendipitous.

### 2.6. Solvent Deuterium Oxide Isotope Effects on Reaction Rate Constants

Analyses of isotope effects in D_2_O relative to H_2_O have been useful in clarifying details of the AChE reaction pathway with a variety of substrates [15,16], but we’ve found few reports of these isotope effects on the reactions of phosphorylated AChE. In our experiments in this report, we paired rate constant determinations in H_2_O and D_2_O by running them on the same day, starting with the same stock AChE frozen aliquot and employing the same dilution sequence. Our results are presented in Table 1, and they are examined in detail in the Discussion.

## 3. Discussion

### 3.1. Kinetics of Decarbamoylation and of the Hydrolysis of Organophosphorylated AChE

Rate constants for organophosphorylated AChE in Table 1 are in reasonable agreement with corresponding values previously reported in slightly different solvents, pH conditions and temperatures. Values of *k*_A_ = 3000 × 10^−6^ min^−1^, of *k*_H_ = 17,000 × 10^−6^ min^−1^, and of *k*_R_ = 480,000 × 10^−6^ min^−1^ were obtained for dimethylphosphorylated human AChE [17]. For diethylphosphorylated AChE, values of *k*_A_ = 170 × 10^−6^ min^−1^ and of *k*_H_ = 120 × 10^−6^ min^−1^ were obtained with the mouse enzyme [18] and of *k*_R_ = 60,000 × 10^−6^ min^−1^ for the human enzyme [19].

An increase in size of the carbamoyl or organophosphoryl group attached to AChE significantly reduced the value of *k*_3_ or *k*_H_ for deacylation. In Table 1, the *k*_H_ for diethylphosphorylated AChE is about 75-fold lower than that for dimethylphosphorylated AChE. A similar trend was seen with carbamoylated AChE, where *k*_3_ for *N,N*-diethylcarbamoyl AChE was about 300-fold lower than that for *N*,*N*-dimethylcarbamoyl AChE [2]. The difference for the organophosphorylated AChEs may result from distortion of the AChE active site by the larger diethylphosphoryl group as seen in Figure 2A. The organophosphoryl group size appears less important for oxime reactivation, as *k*_R_ in Table 1 for 2-PAM reactivation of diethylphosphorylated AChE is only 5-fold lower than that for dimethylphosphorylated AChE. To address the consequences of active site distortion more directly, we analyzed the solvent D_2_O isotope effects in the following section.

### 3.2. Solvent Deuterium Oxide Isotope Effects on Rate Constants for Reactions of Organophosphorylated AChE

When proton transfer occurs in the rate-limiting step of a reaction, the rate constant for that reaction shows a solvent D_2_O isotope effect. Rate-limiting proton transfer in an acylation or deacylation reaction results in a rate constant decrease of 2 to 3-fold when D_2_O replaces H_2_O as the solvent [20]. AChE-catalyzed acetylcholine hydrolysis falls well within this range, as *k*_cat_ (a combination of acylation and deacylation rate constants) has a solvent D_2_O isotope effect of 2.4 [15,21]. However, the isotope effect drops from 2.4 for *k*_cat_ to 1.1–1.2 for the second order hydrolysis rate constant *k*_cat_/*K*_app_ (also denoted *k*_E_) with good substrates of AChE like acetylcholine. To interpret this drop, we proposed that proton transfer in the acylation step *k*_2_ was rate-limiting for *k*_cat_ but that an earlier step like substrate binding, or an induced-fit conformational change that does not involve proton transfer, was rate-limiting for the second order rate constant *k*_cat_/*K*_app_ [15]. More explicitly, the solvent D_2_O isotope effect is determined by the commitment to catalysis, denoted C [16]. In the very simple case of Scheme 3 and Equation (1), acetylcholine (S) binds to enzyme *E* with an association rate constant of *k*_S_ and a dissociation rate constant of *k*_-S_ to give an *E*S complex.

Subsequent acylation with a rate constant *k*_2_ gives an acyl enzyme and choline products, here just denoted P. The observed rate constant *k*_E_ is given by Equation (1). The acylation step *k*_2_ involves proton transfer and has an isotope effect of 2–3, while *k*_S_ and *k*_−S_ are assumed not to involve proton transfer. The commitment to catalysis is defined as C = *k*_2_/*k*_−S_. When C is small, *k*_E_ = *k*_S_
*k*_2_/*k*_−S_, and *k*_E_ shows an isotope effect. When C is large, *k*_E_ = *k*_S_ and it does not.
(1)$$k_{E}=\frac{k_{S}k_{2}}{\left(k_{-S}+k_{2}\right)}$$

A comparable interpretation may be applied to the aging reaction in Scheme 2. This reaction is enzyme catalyzed [22] and involves loss of an alkyl group as the organophosphoryl adduct is converted from a phosphate triester to a diester. While a proton is transferred to the alkyl leaving group in this reaction, this transfer does not appear to be rate limiting as the solvent D_2_O isotope effects for the aging reaction rate constant *k*_A_ in Table 1 are about 1.0. We have found no other reports of isotope effects on aging of dimethyl- or diethylphosphorylated AChE, but a similar isotope effect of 1.2 has been reported for aging of 2-propoxy-methylphosphonylated AChE [23].

We next asked whether the hydrolysis and oxime reactivation reactions in Scheme 2 involve general acid-base catalysis with rate-limiting proton transfer. To facilitate the analysis, we consider Scheme 4, which shows a slight extension of Scheme 1 in which a second acyl enzyme species is added in conformational equilibrium with the first. In particular, Scheme 4 includes an inactive enzyme form (*E*_1_C or *E*_1_OP) with a distorted active site and an active enzyme form (*E*_2_C or *E*_2_OP) that can undergo hydrolysis with a rate constant *k*_3_ or *k*_H_. The forward and reverse rate constants for these equilibria are *k*_F_ and *k*_-F_, respectively, and they are assumed not to involve proton transfer. The general solution to Scheme 4 is more complicated than that of Scheme 3, as neither *E*_2_C nor *E*_2_OP necessarily is in the steady state. However, two extreme cases may be considered in which these intermediates are in the steady state: (1) the commitment to catalysis *k*_3_/*k*_−__F_ or *k*_H_/*k*_−__F_ is small, and the deacylation rate constant *k*_3_*k*_F_/(*k*_−__F_+*k*_3_) or *k*_H_*k*_F_/(*k*_−__F_+*k*_H_) shows the solvent D_2_O isotope effect inherent in *k*_3_ or *k*_H_; (2) the commitment to catalysis *k*_3_/*k*_−__F_ or *k*_H_/*k*_-F_ is large, the acyl intermediates are not equilibrated and the deacylation rate constant *k*_3_*k*_F_/(*k*_−__F_+*k*_H_) or *k*_H_*k*_F_/(*k*_−__F_+*k*_H_) shows only a fraction of the isotope effect inherent in *k*_3_ or *k*_H_ and no isotope effect if *k*_3_ or *k*_H_ >> *k*_−__F_.

We recently investigated the hydrolysis of carbamoylated AChEs for a series of carbamate esters and obtained the solvent D_2_O isotope effects summarized in Figure 6 [2]. The rate constant *k*_3_ for the smallest carbamoyl group, an *N*-monomethylcarbamoyl, was 12 × 10^−3^ min^−1^, and its isotope effect of 2.8 was consistent with rate-limiting proton transfer and a small value of C. However, as the *N*-alkyl groups on carbamoylated AChEs increased in size, the decarbamoylation rate constant *k*_3_ decreased to about 0.02 × 10^−3^ min^−1^ for *N*,*N*-diethylcarbamoylated AChE and the isotope effect was only slightly above 1 (Figure 6), indicating a high value of C. In that report we noted the crystallographic evidence of active site distortion in diethylphosphorylated AChE shown in Figure 2A, and we suggested that a larger size of the carbamoyl group is likely to be an important factor in a shift away from proton transfer in the rate-limiting step for *k*_3_. It may be useful to explore molecular modeling of the array of conformational variants available in AChEs with large carbamoyl groups to obtain support for this proposal.

We designed the experiments here to examine whether active site distortion in diethylphosphorylated AChE in fact does result in a shift away from rate-limiting proton transfer in the hydrolysis and oxime reactivation reactions of this AChE intermediate. For comparison, we included measurements of solvent D_2_O isotope effects for dimethylphosphorylated AChE, an intermediate unlikely to involve active site distortion [10,12]. Oxime reactivation, like hydrolysis in Scheme 2, involves attack of a nucleophile on the tetravalent phosphorus to presumably form a pentavalent transition state [24]. This state then decomposes to regenerate active enzyme and an organophosphorylated oxime [7,25], and both the formation and decomposition of this state are likely to involve rate-limiting proton transfer. In the structure in Figure 2B, 50 mM 2-PAM was diffused into the crystal [9]. This structure suggests that 2-PAM stabilizes the active *E*_2_OP species in Scheme 4, perhaps to the exclusion of the distorted *E*_1_OP species, and *k*_R_ for diethylphosphorylated AChE in Table 1 is only 5-fold lower than that for dimethylphosphorylated AChE. These observations suggest that the concentration of *E*_1_OP bound to 2-PAM is negligible and that *k*_R_ will reflect the rate-limiting proton transfer inherent when both acyl groups migrate to the oxime. The solvent D_2_O isotope effects of 2.0 and 1.7 for the dimethyl- and diethylphosphorylated AChE are consistent with this interpretation.

Finally, we examined the rate constants *k*_H_ for hydrolysis of both diethylphosphorylated AChE and dimethylphosphorylated AChE to determine if their solvent D_2_O isotope effects reflect the fact that only diethylphosphorylated AChE appears to form the distorted active conformation in Figure 2A. The values in Table 1 do not support this proposal, as both organophosphorylated species show isotope effects that are close to 2.0. This value, although somewhat low for typical enzyme deacylation, suggests that any distorted conformation *E*_1_OP involving diethylphosphorylated AChE equilibrates relatively rapidly with the active conformation *E*_2_OP and that the commitment to catalysis C = *k*_H_/*k*_−F_ is small. These isotope effects for organophosphorylated AChE hydrolysis are significant because this reaction, of those in Scheme 2, bears the closest resemblance to the hydrolysis of the carbamoylated enzymes in Figure 6. Furthermore, the conclusion above that the commitment to catalysis C = *k*_3_/*k*_−F_ is large for *N*,*N*-diethylcarbamoylated AChE while C = *k*_H_/*k*_−F_ is small for diethylphosphorylated AChE is difficult to justify. Their respective rate constants, *k*_3_ = 0.02 × 10^−3^ min^−1^ for decarbamoylation and *k*_H_ = 0.08 × 10^−3^ min^−1^ for dephosphorylation (Table 1 and Figure 6), are too similar to support these opposing conclusions about C if the same value of *k*_-F_ holds for both acylated AChEs.

## 4. Materials and Methods

### 4.1. Reagents

Recombinant hAChE was expressed as a secreted, disulfide-linked dimer in drosophila S2 cells and purified by affinity chromatography as outlined previously [26]. Initial 0.5 mL fractions were maintained in 5 mM decamethonium bromide (Sigma-Aldrich Chemical Co., St. Louis, MO, USA) at 4 °C. Fractions were dialyzed against 20 mM sodium phosphate and 0.02% Triton X-100 (pH 7.0) at 4 °C, and 10- or 20- μL aliquots of the dialyzates were frozen at −20 °C until use. *O*,*O*-Diethyl *O*-(4-nitrophenyl) phosphate (paraoxon) and *O*,*O*-dimethyl *O*-(4-nitrophenyl) phosphate (paraoxon methyl) were commercial samples from Sigma-Aldrich Chemical Co (St. Louis, MO, USA).

### 4.2. Assay of Substrate Hydrolysis

Hydrolysis rates *v* for the substrate acetylthiocholine were measured in a coupled Ellman reaction in which thiocholine generated in the presence of the indicated concentration of DTNB was determined by the formation of the thiolate dianion of DTNB at 412 nm (Δε_412 nm_ = 14,150 M^−1^cm^−1^) [27]. Total AChE concentrations (*E*_tot_) could be calculated assuming 450 units/nmol [28]. One unit of AChE activity corresponds to 1 μmol of acetylthiocholine hydrolyzed/min under standard pH-stat assay conditions at pH 8 [28,29]. Our conventional spectrophotometric assay at 412 nm was conducted in pH 7.0 buffer. With wild type hAChE and 0.5 mM acetylthiocholine, this assay resulted in 4.8 ΔA_412 nm_/min with 1 nM AChE or about 76% of the pH stat assay standard, but here *E*_tot_ was expressed simply in terms of ΔA_412_/min. The assay mixture here contained final concentrations of acetylthiocholine and DTNB of 1.0 mM and 0.3 mM, respectively, in 100 mM sodium phosphate and 0.1% BSA at pH 8.0. Aliquots of AChE were added to a final volume was 3.0 mL, and the assay was conducted at 25 °C. Absorbance at 412 nm was recorded with time on a Varian Cary 3A spectrophotometer.

### 4.3. Reactions of Organophosphorylated hAChE

We generated organophosphorylated AChE (*E*OP in Scheme 1) from either paraoxon (to give diethylphosphorylated AChE) or paraoxon methyl (to give dimethylphosphorylated AChE) as outlined in the legends to Figure 3, Figure 4 and Figure 5. Experimental traces of organophosphorylated AChE reactions measured with acetylthiocholine activity assays (*v*) were obtained and the equation for the general solution used to fit these traces to Scheme 2 is given in Equation (2). Data were fitted to Equation (2) by unweighted nonlinear regression with SigmaPlot (version 12.0, Systat Software Inc., Chicago, IL, USA). When *k*_r_ >> *k*_H_ + *k*_A_ + *k*_X_ and *t* = *t*_aged_ >> *k*_r_^−1^, Equation (2) reduces to Equation (3).
(2)$$v=\left(E_{\mathrm{tot}}e^{-k_{A}t_{\mathrm{aged}}}-E_{0}\right)\frac{\left(k_{r}+k_{H}\right){\left(e^{-k_{X}t}-e^{-\left(k_{r}+k_{H}+k_{A}+k_{X}\right)t}\right)}^{}}{\left(k_{r}+k_{H}+k_{A}\right)}+E_{0}\;e^{-k_{X}t}$$
(3)$$v=E_{\mathrm{tot}}e^{-\left(k_{A}+k_{X}\right)t_{\mathrm{aged}}}$$

In Equations (2) and (3), *t*_aged_ is the time that enzyme and OP were incubated prior to the start of the measured reaction, *t* is the duration in min of the measured reaction and *E*_0_ is the initial enzyme concentration (ΔA_412_/min at *t* = 0).

Values of the second order reactivation rate constants *k*_r_ were obtained at various 2-PAM concentrations that were run in parallel, and these values were analyzed with Equation (4) to obtain the maximal first order reactivation rate constant *k*_R_ at saturating concentrations of 2-PAM and the dissociation constant *K*_P_ for 2-PAM binding to organophosphorylated AChE.
(4)$$k_{r}=\frac{k_{R}}{1+\frac{K_{P}}{\left[2-\mathrm{PAM}\right]}}$$

### 4.4. Solvent Deuterium Oxide Isotope Effects

Reactions of organophosphorylated AChE were conducted in 100 mM sodium phosphate and 0.1% BSA at pH 8.0. The pH was the uncorrected value read by the pH meter for both H_2_O and D_2_O buffers, and the pH of 8.0 (rather than more typical pH values of 7.0–7.4 for AChE studies) was selected to minimize enzyme pK_a_ differences [3] between H_2_O and D_2_O. Organophosphorylated AChE reactions in D_2_O were paired with reactions in H_2_O to maximize precision in comparing rates. While frozen stocks of AChE were in H_2_O, all subsequent dilutions were identical in either H_2_O or D_2_O. The percentage H_2_O in an assayed D_2_O aging reaction mixture was 3%, and the maximum percentage in hydrolysis or oxime reactivation mixtures was 1%.

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
