# Peer review of "Solvent Deuterium Oxide Isotope Effects on the Reactions of Organophosphorylated Acetylcholinesterase [Author-notes fn1-molecules-25-04412]"

_molecules, 2020, doi:10.3390/molecules25194412_

Round 1

Reviewer 1 Report

To go further with the publishing process authors should do following:

(1) Make uniform all the schemes, it seems that half of schemes were drawn, half of the copy-pasted

(2) Perform molecular modeling (DFT or docking) to support/conform experimental findings  

Reviewer 2 Report

The presented paper brings interesting work worth publishing; however, it needs a major revision to be acceptable.

General comments:

A major part of the Introduction section is copied from author’s previously published paper in Chemico-Biological Interactions 308, 2019., where it was a part of discussion and this paper has not been cited in this manuscript. I do understand that you cannot say much differently what AChE is (line 34-38) but describing results from crystallographic studies (lines 53-64; 71-78) could be rephrased or just referred to. Also, Figure 2 with its description is exactly the same as published in the CBI paper mentioned above which is not stated. The figure itself is important but I do not know whether copyrights are in question.

Furthermore, the MS is divided in Results section followed by Discussion. However, Results obtained are not clearly presented, or in other words it is not clear what the author wanted to point as the main results (with the exception of the ones described in 2.1. subtitle). Namely, a lot of accent is given to discussion and description of how the results were obtained and constants calculated, leaving the reader questioning what the main results were in the end? I suggest therefore clearing this section and describing the numbers not protocols. If the main results were obtaining the right fit with the proposed equations than this needs to be pointed out.

Furthermore, there is confusion with figures and schemes numbering which makes results for a reader difficult to follow. This is probably, I guess, a consequence of moving Materials and Methods section to the end, due to the propositions of the Journal, but it needs to be sorted out. In example, now you have mentioning of Fig. 4 and 5 before Fig 3., or describing Scheme 2 that is presented after schemes 3 and 4 later in the manuscript. As the Guide for authors says: “all Figures, Schemes and Tables should be inserted into the main text close to their first citation and must be numbered following their number of appearance. All Figures, Schemes and Tables should have a short explanatory title and caption”… which is missing now for all the schemes. Furthermore, the figures and schemes description should bear only what is presented in the graphs and perhaps fitting info, while all the details on how the data was experimentally obtained should be written in the Materials and Method section. Therefore, all the procedures for measuring, that are in detail described under figure titles, should be transferred to M&M where at the moment none of it is present.

Do not know have I missed the data in reading, but, either section Results or Discussion is missing a comparison of the obtained constants (for H2O solvent experiments primarily) with the ones previously published in the literature, especially for aging and reactivation of POX with 2-PAM, just to verify all constants calculated with here stated procedure.

Specific comments:

  1. Abstract needs to be modified to address the results presented in this paper not results from previously published work. Namely, references should be omitted and discussion in terms of going to details like numbers (line 21-24) about decarbamoylation should be shortened and results of this research should be pointed out giving perhaps one or two more sentences. I suggest shortening also the initial part of the abstract like (since this is for the Special issue):

“Organophosphates (OPs) are esters of substituted phosphoric acids, used as pesticides and nerve agents, that react with acetylcholinesterase (AChE) by initially transferring the organophosphoryl group to a serine residue in the enzyme active site, concomitant with loss of an alcohol leaving group”….

In that sense the sentence in lines 15-17 (“OPs thus are very effective….) should be removed.

  1. I suggest modifying/shortening keywords and not to repeat the title words to be something like: cholinesterases, reactivation, paraoxon, D2O isotope, dephosphorylation or something similar.
  2. In the Introduction part line 42 author states that substrate does not form reversible complex with the enzyme while later in the discussion refers to the general scheme of substrate hydrolysis where this complex is indicated (and also in the text, line 250, complex is mentioned). As I am aware, this complex is supposed to be formed, so the author needs to state or explain to what this refers to or what was intended to be said?
  3. There is naming of POX-E complex as diethylphosphorylated and diethoxyphosphorylated, so this should be uniformed. I lean to the use of diethoxyphosphorylated/dimethoxyhosphorylated just to make a difference from the carbamoyl substituents used in the text for comparison. Also, to be exactly right I thing that the proper perhaps should be O-diethylphosphorylated or O-dimethoxyhosphorylated. Anyhow, it just needs to be uniformed throughout the text.
  4. Reference is missing at the end of the sentence in line 89-92 referring to the methyl-POX crystal structure?
  5. Figure 1. - this figure is somehow unnecessary (quite similar is presented in scheme 2 and table 2) or if the author choose to be included, it needs more details to be clearer for a wider audience like names of these compounds attached. This is a general structure presentation that looks exactly the same for all species active Ser bound complex, and naming it as “…substituents at hAChE Ser203….” is not appropriate, especially if you do not put Ser203 mark in the picture. Furthermore, these substituents are just examples of many more, so I suggest putting in the figure title something like “Molecular structures of several....”.
  6. Results presented in Figures 3 and 5 are given only for methyl-POX while in Figure 4 for POX. It would be better to see graphs for both OPs or to say clearly that these are just representative experiments (since also experimental errors are missing in most cases). Also, it would be nice to see a comparison of experiments in H2O and D2O, not just final reactivation curves as in Figure 5.
  7. Figure 3 – subtitles should be described under the figure legend not in the figures itself since there is already A/B numbering. Under A panel, units - “min” (or the appropriate one) - should be given to legend numbers 1, 180, 420 and 1470.
  8. Figure 4 title is referring only to hydrolysis while also denaturation of the enzyme is given. I would suggest maybe putting two panels and/or renaming figure title. Also graph legend given under blue and green should be described in the figure description and only short naming given in the graph itself. Written like it is, it is not clear for presentation.
  9. Table 1 – the values for ks obtained for D2O should be given as well with corresponding n (not just ratio H20/D2O). I understand that n=2 will be sufficient if the two experiments errors are minimal, but why for aging for diethyl-POX n=1? I do not think that you can draw a conclusions based on only one experiment, especially if you do not perhaps cite other research that had obtained the same value so this would be only like a confirmation of previously published results. Furthermore, marking the “k” as “k*106” in the first line is confusing. I guess written like that means you have to recalculate your exact k values by dividing numbers with 106, but that can also be misunderstood and present possible misquotations for future papers, especially for younger audience. I would suggesting using only k (10-6 min-1).
  10. Table 2 has been published in two previous papers from the author so I do not see why it has to be included here as a separate table again, and not just citing data in the text? Also, Table 2 is repeating a part of Figure 1 which is unnecessary. If the author wants for comparison to have these numbers explicitly stated they should be combined in Table 1 and adequately cites.
  11. Schemes need to be uniformed in naming of reactions substrates and products to make them completely understandable and connected. Namely, Schemes 1 bring E + COH or POH while Scheme 3 brings only “P” as a product where analogy should be E + P or in the Schemes 4 there is also only “P” where E + COH or E + POH should be repeated.
  12. Subtitle 3.1. needs to be changed to “kinetics of dephosphorylation….” not to say “kinetics of decarbamoylation….” to be in consistency with the manuscript title.
  13. Reactivation rate constant kr and kR should be named differently into “observed first-order reactivation rate constants” and “maximal first-order reactivation rate constants”, respectively, not to be confused with the overall second order reactivation rate constant for this reaction. Namely, this is the only reaction followed in the manuscript that depends on the two substrates concentrations entering reaction - OP-enzyme and the oxime - (if we ignore water in hydrolysis) so it should be stated directly somehow in descriptions and oxime (and water perhaps) should be added in the Scheme 2 d) (or c)) under the arrow.
  14. There has been some “template error” at page 10 where the manuscript text ended up under Scheme 4 legend, and it seems like there are several extra spaces separations throughout the text before beginning of the sentences, as well as unequal line spacing (page 10 and 11).

Reviewer 3 Report

This manuscript comes from one of pioneers and strongest contributors to the field of cholinesterase (ChE) reaction kinetics and that per se deserves particular attention in evaluating its content. As noted by the author, deuterium oxide isotope effects are a valuable tool in discerning details of reaction mechanisms and while it has been used in studying catalytic mechanisms of native cholinesterases in the past, reactions involving covalently inhibited ChEs, organophospahate (OP) or carbamate (CE) covalent ChE conjugates, were less of a topic of studies involving isotope effects. Experiments described in this manuscript contribute significantly to the insight into structure/activity of ChEs, in particular those of human acetylcholinesterase (hAChE). The report is based on the previously published X-ray structure observations that diethylphosphorylated hChE unlike dimethylphosphorylated AChE shows significant distortion of the loop in the vicinity of the reactive active serine, that consequently could affect kinetics and rate limiting steps in three distinct reactions of OP-hAChE: spontaneous deposphorylation, enzyme-catalyzed dealkylation (aging) and oxime induced deposphorylation (oxime reactivation). The hypothesis was suggested that solvent isotope effects should indicate potential difference in rate limiting steps of the two OP-hAChE conjugates, due to backbone distortion in one of them. The outcome was that no difference was observed, and that is a significant finding in potential contradiction with intuition. That may promote more research of similar kind in the field, that will look for specific structural basis, related to the absence of the effect.

Several smaller corrections in the manuscript could be suggested, including solicitation for some additional comments. However, a part of the text seems to be missing (technical issue?) calling for a more substantial revision.

Major issues:

  1. The text abruptly breaks in line 268 with no continuation on the topic in line 269 or anywhere further.
  2. The study is on organophosphoRYLated hAChE, however in general descriptions (abstract, introduction) of OP reactions with ChEs substituted phosphonates and phosphoramidates should also be specifically mentioned. Those OPs are largely chemical warfare agents and not phosphoryc acid esthers, that are mainly used as pesticides.
  3. Throughout the text “aging” is described as a “…loss of a second alcohol group …” while for most cases literature in the field describes aging as dealkylation or loss of an alkyl group that has been demonstrated as a dominant aging mechanism and that implies some difference in the reaction mechanism of “aging”.
  4. While value of Kp=170uM was given for the oxime reactivation reaction in H2O, one was unable to find an analogous constant for reaction in D2O, that obviously determined in the non-linear fit. Would the author be open to a possible interpretation that the difference in Kp could “mask” the isotope effect for the diethylphosphorylated hAChE?
  5. Please align order/numbering of figures and schemes with sequence of their first mention in the text.

Minor issues:

  1. The line 21 mentiones pralidoxime. An analogous structure apparently exists for another oxime, HI-6, with similar effect on the main fold of the hAChE.
  2. Lines 43-45 describe reversible complex formation in the course of OP conjugation as a difference from acetylcholine hydrolytic mechanism. Hasn’t acetylcholine been demonstrated to form an analogous reversible complex characterized by a Ks constant (as opposed to Km), both kinetically and crystallographically?
  3. Table 1: It is not incorrect, but representation of units for “k” might be easier to follow in the format k (10^6 min-1).
  4. Lines 302 and 303: at what concentration of 2-PAM did it stabilize the conformation?

Round 2

Reviewer 2 Report

The MS sounds better now and by the addition of several sentences, explanations, and sorting the schemes it is easier to follow. I have noticed only a minor mistake regarding Scheme 2 naming font size (...c)hydrolysis - is several sizes bigger) but i guess this could be modified in later copyediting process.

I do agree with the author that it is not for a reviewer to rewrite the MS, but if the data is not as clearly presented as expected, and therefore the text is difficult to follow, every objective suggestion for improvement should be welcomed and considered, not judged.

Anyhow, the MS is now in my opinion suitable for publication.